# Meta-Curvature

**Eunbyung Park**
Department of Computer Science
University of North Carolina at Chapel Hill
eunbyung@cs.unc.edu

**Junier B. Oliva**
Department of Computer Science
University of North Carolina at Chapel Hill
joliva@cs.unc.edu

## Abstract

We propose *meta-curvature* (MC), a framework to learn curvature information for better generalization and fast model adaptation. MC expands on the model-agnostic meta-learner (MAML) by learning to transform the gradients in the inner optimization such that the transformed gradients achieve better generalization performance to a new task. For training large scale neural networks, we decompose the curvature matrix into smaller matrices in a novel scheme where we capture the dependencies of the model's parameters with a series of tensor products. We demonstrate the effects of our proposed method on several few-shot learning tasks and datasets. Without any task specific techniques and architectures, the proposed method achieves substantial improvement upon previous MAML variants and outperforms the recent state-of-the-art methods. Furthermore, we observe faster convergence rates of the meta-training process. Finally, we present an analysis that explains better generalization performance with the meta-trained curvature.

## 1   Introduction

Despite huge progress in artificial intelligence, the ability to quickly learn from few examples is still far short of that of a human. We are capable of utilizing prior knowledge from past experiences to efficiently learn new concepts or skills. With the goal of building machines with this capability, *learning-to-learn* or *meta-learning* has begun to emerge with promising results.

One notable example is model-agnostic meta-learning (MAML) [9, 30], which has shown its effectiveness on various few-shot learning tasks. It formalizes *learning-to-learn* as meta objective function and optimizes it with respect to a model's initial parameters. Through the meta-training procedure, the resulting model's initial parameters become a very good prior representation and the model can quickly adapt to new tasks or skills through one or more gradient steps with a few examples. Although this end-to-end approach, using standard gradient descent as the inner optimization algorithm, was theoretically shown to approximate any learning algorithm [10], recent studies indicate that the choice of the inner-loop optimization algorithm affects performance. [22, 4, 13].

Given the sensitivity to the inner-loop optimization algorithm, second order optimization methods (or preconditioning the gradients) are worth considering. They have been extensively studied and have shown their practical benefits in terms of faster convergence rates [31], an important aspect of few-shot learning. In addition, the problems of computational and spatial complexity for training deep networks can be effectively handled thanks to recent approximation techniques [24, 38]. Nevertheless, there are issues with using second order methods in its current form as an inner loop optimizer in the meta-learning framework. First, they do not usually consider generalization performance. They compute local curvatures with training losses and move along the curvatures as far as possible. It can be very harmful, especially in the few-shot learning setup, because it can overfit easily and quickly.

---

The code is available at `https://github.com/silverbottlep/meta_curvature`

In this work, we propose to learn a curvature for better generalization and faster model adaptation in the meta-learning framework, we call *meta-curvature*. The key intuition behind MAML is that there are some representations are broadly applicable to all tasks. In the same spirit, we hypothesize that there are some curvatures that are broadly applicable to many tasks. Curvatures are determined by the model's parameters, network architectures, loss functions, and training data. Assuming new tasks are distributed from the similar distribution as meta-training distribution, there may exist common curvatures that can be obtained through meta-training procedure. The resulting meta-curvatures, coupled with the simultaneously meta-trained model's initial parameters, will transform the gradients such that the updated model has better performance on new tasks with fewer gradient steps. In order to efficiently capture the dependencies between all gradient coordinates for large networks, we design a multilinear mapping consisting of a series of tensor-products to transform the gradients. It also considers layer specific structures, e.g. convolutional layers, to effectively reflects our inductive bias. In addition, meta-curvature can be easily implemented (simply transform the gradients right before passing through the optimizers) and can be plugged into existing meta-learning frameworks like MAML without additional, burdensome higher-order gradients.

We demonstrate the effectiveness of our proposed method on the few-shot learning tasks done by [44, 34, 9]. We evaluated our methods on few-shot regression and few-shot classification tasks over Omniglot [19], miniImagenet [44], and tieredImagnet [35] datasets. Experimental results show significant improvements on other MAML variants on all few-shot learning tasks. In addition, MC's simple gradient transformation outperformed other more complicated state-of-the-art methods that include additional bells and whistles.

## 2   Background

### 2.1   Tensor Algebra

We review basics of tensor algebra that will be used to formalize the proposed method. We refer the reader to [17] for a more comprehensive review. Throughout the paper, tensors are defined as multi-dimensional arrays and denoted by calligraphic letters, e.g. $N$th-order tensor, $\mathcal{X} \in \mathbb{R}^{I_1 \times I_2 \times \cdots \times I_N}$. Matrices are second-order tensors and denoted by boldface uppercase, e.g. $\mathbf{X} \in \mathbb{R}^{I_1 \times I_2}$.

**Fibers:** Fibers are a higher-order generalization of matrix rows and columns. A matrix column is a mode-1 fiber and a matrix row is a mode-2 fiber. The mode-1 fibers of a third order tensor $\mathcal{X}$ are denoted as $\mathcal{X}_{:,j,k}$, where a colon is used to denote all elements of a mode.

**Tensor unfolding:** Also known as *flattening (reshaping)* or *matricization*, is the operation of arranging the elements of an higher-order tensors into a matrix. The mode-$n$ unfolding of a $N$th-order tensor $\mathcal{X} \in \mathbb{R}^{I_1 \times I_2 \times \cdots \times I_N}$, arranges the mode-n fibers to be the columns of the matrix, denoted by $\mathcal{X}_{[n]} \in \mathbb{R}^{I_n \times I_M}$, where $I_M = \prod_{k \neq n} I_k$. The elements of the tensor, $\mathcal{X}_{i_1,i_2,\dots,i_N}$ are mapped to $\mathcal{X}_{[n]i_n,j}$, where $j = 1 + \sum_{k \neq n, k=1}^{N}(i_k - 1)J_k$, with $J_k = \prod_{m=1, m\neq n}^{k-1} I_m$.

**$n$-mode product:** It defines the product between tensors and matrices. The $n$-mode product of a tensor $\mathcal{X} \in \mathbb{R}^{I_1 \times I_2 \times \cdots \times I_N}$ with a matrix $\mathbf{M} \in \mathbb{R}^{J \times I_n}$ is denoted by $\mathcal{X} \times_n \mathbf{M}$ and computed as

$$(\mathcal{X} \times_n \mathbf{M})_{i_1,\dots,i_{n-1},j,i_{n+1},\dots,i_N} = \sum_{i_n=1}^{I_n} \mathcal{X}_{i_1,i_2,\dots,i_N} \mathbf{M}_{j,i_n}. \tag{1}$$

More concisely, it can be written as $(\mathcal{X} \times_n \mathbf{M})_{[n]} = \mathbf{M}\mathcal{X}_{[n]} \in \mathbb{R}^{I_1 \times \cdots \times I_{n-1} \times J \times I_{n+1} \times \cdots \times I_N}$. Despite cumbersome notation, it is simply $n$-mode unfolding (reshaping) followed by matrix multiplication.

### 2.2   Model-Agnostic Meta-Learning (MAML)

MAML aims to find a transferable initialization (a prior representation) of any model such that the model can adapt quickly from the initialization and produce good generalization performance on new tasks. The meta-objective is defined as validation performance after one or few step gradient updates from the model's initial parameters. By using gradient descent algorithms to optimize the meta-objective, its training algorithm usually takes the form of nested gradient updates: inner updates for model adaptation to a task and outer-updates for the model's initialization parameters. Formally,

$$\min_{\theta} \mathbb{E}_{\tau_i}[\mathcal{L}_{\text{val}}^{\tau_i}(\underbrace{\theta - \alpha \nabla \mathcal{L}_{\text{tr}}^{\tau_i}(\theta)}_{\text{inner udpate}})], \tag{2}$$

where $\mathcal{L}_{\text{val}}^{\tau_i}(\cdot)$ denotes a loss function for a validation set of a task $\tau_i$, and $\mathcal{L}_{\text{tr}}^{\tau_i}(\cdot)$ for a training set, or $\mathcal{L}_{\text{tr}}(\cdot)$ for brevity. The inner update is defined as a standard gradient descent with fixed learning rate $\alpha$. For conciseness, we assume as single adaptation step, but it can be easily extended to more steps. For more details, we refer to [9]. Several variations of inner update rules were suggested. Meta-SGD [22] suggested coordinate-wise learning rates, $\theta - \alpha \circ \nabla \mathcal{L}_{\text{tr}}$, where $\alpha$ is the learnable parameters and $\circ$ is element wise product. Recently, [4] proposed a learnable learning rate per each layers for more flexible model adaptation. To alleviate computational complexity, [30] suggested an algorithm that do not require higher order gradients.

## 2.3 Second order optimization

The biggest motivation of second order methods is that first-order optimization such as standard gradient descent performs poorly if the Hessian of a loss function is ill-conditioned, e.g. a long narrow valley loss surface. There are a plethora of works that try to accelerate gradient descent by considering local curvatures. Most notably, the update rules of Newton's method can be written as $\theta - \alpha \mathbf{H}^{-1} \nabla \mathcal{L}_{\text{tr}}$, with Hessian matrix $\mathbf{H}$ and a step size $\alpha$ [31]. Every step, it minimizes a local quadratic approximation of a loss function, and the local curvature is encoded in the Hessian matrix. Another promising approach, especially in neural network literature, is natural gradient descent [2]. It finds a steepest descent direction in distribution space rather than parameter space by measuring KL-divergence as a distance metric. Similar to Newton's method, it preconditions the gradient with the Fisher information matrix and a common update rule is $\theta - \alpha \mathbf{F}^{-1} \nabla \mathcal{L}_{\text{tr}}$. In order to mitigate computational and spatial issues for large scale problems, several approximation techniques has been proposed, such as online update methods [31, 38], Kronecker-factored approximations [24], and diagonal approximations of second order matrices [43, 16, 8].

# 3 Meta-Curvature

We propose to learn a curvature along with the model's initial parameters simultaneously via the meta-learning process. The goal is that the meta-learned curvature works collaboratively with the meta-learned model's initial parameters to produce good generalization performance on new tasks with fewer gradient steps. In this work, we focus on learning a meta-curvature and its efficient forms to scale large networks. We follow the meta-training algorithms suggested in [9] and the proposed method can be easily plugged in.

## 3.1 Motivation

We begin with the hypothesis that there are broadly applicable curvatures to many tasks. In training a neural network with a loss function, local curvatures are determined by the model's parameters, the network architecture, the loss function, and training data. Since new tasks are sampled from the same or similar distributions and all other factors are fixed, it is intuitive idea that there may exist some curvatures found via meta-training that can be effectively applied to the new tasks. Throughout the meta-training, we can observe how the gradients affect the validation performance and use those experiences to learn how to transform or correct the gradient from the new task.

We take a learning approach because existing curvature estimations do not consider generalization performance, e.g. Hessian and the Fisher-information matrix. The local curvatures are approximated with only current training data and loss functions. Therefore, these methods may end up converging fast to a poor local minimum. This is especially true when we have few training examples.

## 3.2 Method

First, we present a simple and efficient form of the meta-curvature computation through the lens of tensor algebra. Then, we present a matrix-vector product view to provide intuitive idea of the connection to the second order matrices. Lastly, we discuss the relationships to other methods.

### 3.2.1 Tensor product view

We consider neural networks as our models. With a slight abuse of notation, let the model's parameters $\mathcal{W}^l \in \mathbb{R}^{C_{out}^l \times C_{in}^l \times d^l}$ and its gradients of loss function $\mathcal{G}^l \in \mathbb{R}^{C_{out}^l \times C_{in}^l \times d^l}$, at each layers $l$. To avoid

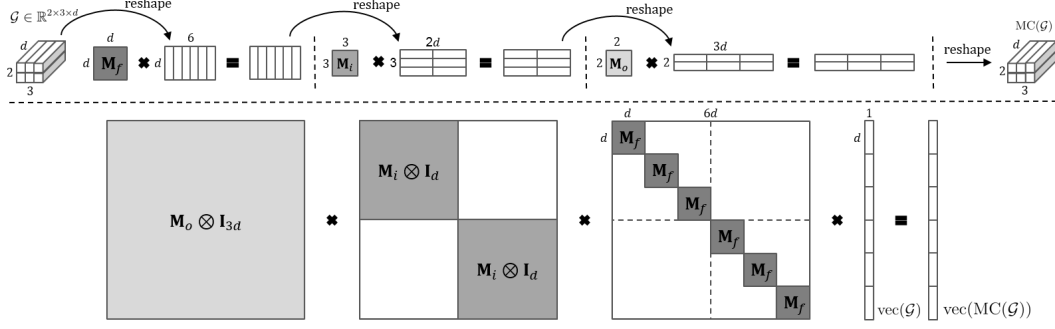

Figure 1: An example of meta-curvature computational illustration with $\mathcal{G} \in \mathbb{R}^{2 \times 3 \times d}$. Top: tensor algebra view, Bottom: matrix-vector product view.

cluttered notation, we will omit the superscript $l$. We choose superscripts and dimensions with 2D convolutional layers in mind, but the method can be easily extended to higher dimension convolutional layers or other layers that consists of higher dimension parameters. $C_{out}, C_{in}$, and $d$ are the number of output channels, the number of input channels, and the filter size respectively. $d$ is height $\times$ width in convolutional layers and 1 in fully connected layers. We also define meta-curvature matrices, $\mathbf{M}_o \in \mathbb{R}^{C_{out} \times C_{out}}$, $\mathbf{M}_i \in \mathbb{R}^{C_{in} \times C_{in}}$, and $\mathbf{M}_f \in \mathbb{R}^{d \times d}$. Now a meta-curvature function takes a multidimensional tensor as an input and has all meta-curvature matrices as learnable parameters:

$$\text{MC}(\mathcal{G}) = \mathcal{G} \times_3 \mathbf{M}_f \times_2 \mathbf{M}_i \times_1 \mathbf{M}_o. \tag{3}$$

Figure 1 (top) shows an example of computational illustration with an input tensor $\mathcal{G} \in \mathbb{R}^{2 \times 3 \times d}$. First, it performs linear transformations for all 3-mode fibers of $\mathcal{G}$. In other words, $\mathbf{M}_f$ captures the parameter dependencies between the elements within a 3-mode fiber, e.g. all gradient elements in a channel of a convolutional filter. Secondly, the 2-mode product models the dependencies between 3-mode fibers computed from the previous stage. All 3-mode fibers are updated by linear combinations of other 3-mode fibers belonging to the same output channel (linear combinations of 3-mode fibers in a convolutional filter). Finally, the 1-mode product is performed in order to model the dependencies between the gradients of all convolutional filters. Similarly, the gradients of all convolutional filters are updated by linear combinations of gradients of other convolutional filters.

A useful property of $n$-mode products is the fact that the order of the multiplications is irrelevant for distinct modes in a series of multiplications. For example, $\mathcal{G} \times_3 \mathbf{M}_f \times_2 \mathbf{M}_i \times_1 \mathbf{M}_o = \mathcal{G} \times_1 \mathbf{M}_o \times_2 \mathbf{M}_i \times_3 \mathbf{M}_f$. Thus, the proposed method indeed examines the dependencies of the elements in the gradient all together.

### 3.2.2 Matrix-vector product view

We can also view the proposed meta-curvature computation as a matrix-vector product analogous to that from other second order methods. Note that this is for the purpose of intuitive illustration and we cannot compute or maintain this large matrices for large deep networks. We can expand the meta-curvature matrices as follows.

$$\widehat{\mathbf{M}_o} = \mathbf{M}_o \otimes \mathbf{I}_{C_{in}} \otimes \mathbf{I}_d, \quad \widehat{\mathbf{M}_i} = \mathbf{I}_{C_{out}} \otimes \mathbf{M}_i \otimes \mathbf{I}_d, \quad \widehat{\mathbf{M}_f} = \mathbf{I}_{C_{out}} \otimes \mathbf{I}_{C_{in}} \otimes \mathbf{M}_f, \tag{4}$$

where $\otimes$ is the Kronecker product, $\mathbf{I}_k$ is $k$ dimensional identity matrix, and the three expanded matrices are all same size $\widehat{\mathbf{M}_o}, \widehat{\mathbf{M}_i}, \widehat{\mathbf{M}_f} \in \mathbb{R}^{C_{out}C_{in}d \times C_{out}C_{in}d}$. Now we can transform the gradients with the meta-curvature as

$$\text{vec}(\text{MC}(\mathcal{G})) = \mathbf{M}_{mc} \text{vec}(\mathcal{G}), \tag{5}$$

where $\mathbf{M}_{mc} = \widehat{\mathbf{M}_o}\widehat{\mathbf{M}_i}\widehat{\mathbf{M}_f}$. The expanded matrices satisfy commutative property, e.g. $\widehat{\mathbf{M}_o}\widehat{\mathbf{M}_i}\widehat{\mathbf{M}_f} = \widehat{\mathbf{M}_f}\widehat{\mathbf{M}_i}\widehat{\mathbf{M}_o}$, as shown in the previous section. Thus, $\mathbf{M}_{mc}$ models the dependencies of the model parameters all together. Note that we can also write $\mathbf{M}_{mc} = \mathbf{M}_o \otimes \mathbf{M}_i \otimes \mathbf{M}_f$, but this is non-commutative, $\mathbf{M}_o \otimes \mathbf{M}_i \otimes \mathbf{M}_f \neq \mathbf{M}_f \otimes \mathbf{M}_i \otimes \mathbf{M}_o$.

Figure 1 (bottom) shows a computational illustration. $\widehat{\mathbf{M}_f}\text{vec}(\mathcal{G})$, which is equivalent computation to $\mathcal{G} \times_3 \mathbf{M}_f$, can be interpreted as a giant matrix-vector multiplication with block diagonal matrix, where

each block shares same meta-curvature matrix $\mathbf{M}_f$. It resembles the block diagonal approximation strategies in some second-order methods for training deep networks, but as we are interested in learning meta-curvature matrices, no approximation is involved. And matrix-vector product with $\widehat{\mathbf{M}_o}$ and $\widehat{\mathbf{M}_i}$ are used to capture inter-parameter dependencies and are computationally equivalent to 2-mode and 3-mode products of Eq. 3.

### 3.2.3 Relationship to other methods

Tucker decomposition [17] decomposes a tensor into low rank cores with projection factors and aims to closely reconstruct the original tensor. We maintain full rank gradient tensors, however, and our main goal is to transform the gradients for better generalization. [18] proposed to learn the projection factors in Tucker decomposition for fully connected layers in deep networks. Again, their goal was to find the low rank approximations of fully connected layers for saving computational and spatial cost.

Kronecker-factored Approximate Curvature (K-FAC) [24, 14] approximates the Fisher matrix by the Kronecker product, e.g. $\mathbf{F} \approx \mathbf{A} \otimes \mathbf{G}$, where $\mathbf{A}$ is computed from the activation of input units and $\mathbf{G}$ is computed from the gradient of output units. Its main goal is to approximate the Fisher such that matrix vector products between its inversion and the gradient can be computed efficiently. However, we found that maintaining $\mathbf{A} \in \mathbb{R}^{C_{in}d \times C_{in}d}$ was quite expensive both computationally and spatially even for smaller networks. In addition, when we applied this factorization scheme to meta-curvature, it tends to easily overfit to meta-training set. On the contrary, we maintain two separated matrices, $\mathbf{M}_i \in \mathbb{R}^{C_{in} \times C_{in}}$ and $\mathbf{M}_f \in \mathbb{R}^{d \times d}$, which allows us to avoid overfitting and heavy computation. More importantly, we learn meta-curvature matrices to improve generalization instead of directly computing them from the activation and the gradient of training loss. Also, we do not require expensive matrix inversions.

### 3.2.4 Meta-training

We follow a typical meta-training algorithm and initialize all meta-curvature matrices as identity matrices so that the gradients do not change at the beginning. We used the ADAM [16] optimizer for the outer loop optimization and update the model's initial parameters and meta-curvatures simultaneously. We provide the details of algorithm in appendices.

## 4 Analysis

In this section, we will explore how a meta-trained matrix $\mathbf{M}_{mc}$, or $\mathbf{M}$ for brevity, can operate for better generalization. Let us take the gradient of meta-objective w.r.t $\mathbf{M}$ for a task $\tau_i$. With the inner update rule $\theta^{\tau_i}(\mathbf{M}) = \theta - \alpha \mathbf{M} \nabla_\theta \mathcal{L}_{\text{tr}}^{\tau_i}(\theta)$, and by applying chain rule,

$$\nabla_\mathbf{M} \mathcal{L}_{\text{val}}^{\tau_i}(\theta^{\tau_i}(\mathbf{M})) = -\alpha \nabla_{\theta^{\tau_i}} \mathcal{L}_{\text{val}}^{\tau_i}(\theta^{\tau_i}) \nabla_\theta \mathcal{L}_{\text{tr}}^{\tau_i}(\theta)^\top, \tag{6}$$

where $\theta^{\tau_i}$ is the parameter for the task $\tau_i$ after the inner update. It is the outer product between the gradients of validation loss and training loss. Note that there is a significant connection to the Fisher information matrix. For a task $\tau_i$, if we define the loss function as negative log likelihood, e.g. a supervised classification task $\mathcal{L}^{\tau_i}(\theta) = \mathbb{E}_{(x,y) \sim p(\tau_i)}[-\log_\theta p(y|x)]$, then the empirical Fisher can be defined as $\mathbf{F} = \mathbb{E}_{(x,y) \sim p(\tau_i)}[\nabla_\theta \log_\theta p(y|x) \nabla_\theta \log_\theta p(y|x)^\top]$. There are three clear distinctions. First, the training and validation sets are treated separately in the meta-gradient $\nabla_\mathbf{M} \mathcal{L}_{\text{val}}^{\tau_i}$, while the empirical Fisher is computed with only training set (validation set is not available during training). Secondly, the gradient of the validation set is evaluated at new parameters $\theta^{\tau_i}$ after the inner update in the meta-gradient. Finally, the Fisher is positive semi-definite by construction, but it is not the case for the meta-gradient. This is an attractive property since it guarantees that the transformed gradient is always a descent direction. However, we mainly care about generalization performance in this work. Hence, we rather not force this property in this work, but leave it for future work.

Now let us consider what the meta-gradient can do for good generalization performance. Given a fixed point $\theta$ and a meta training set $\mathcal{T} = \{\tau_i\}$, standard gradient descent from an initialization $\mathbf{M}$, gives the following update.

$$\mathbf{M}_\mathcal{T} = \mathbf{M} - \beta \sum_{i=1}^{|\mathcal{T}|} \nabla_\mathbf{M} \mathcal{L}_{\text{val}}^{\tau_i}(\theta^{\tau_i}(\mathbf{M})) = \mathbf{M} + \alpha\beta \sum_{i=1}^{|\mathcal{T}|} \nabla_\theta \mathcal{L}_{\text{val}}^{\tau_i}(\theta^{\tau_i}(\mathbf{M})) \nabla_\theta \mathcal{L}_{\text{tr}}^{\tau_i}(\theta)^\top, \tag{7}$$

where $\alpha$ and $\beta$ are fixed inner/outer learning rates respectively. Here, we assume a standard gradient descent for simplicity. But the argument extends to other advanced gradient algorithms, such as momentum and ADAM.

We apply $\mathbf{M}_{\mathcal{T}}$ to the gradients of a new task, giving the transformed gradients

$$\mathbf{M}_{\mathcal{T}} \nabla_\theta \mathcal{L}_{\text{tr}}^{\mathcal{T}_{\text{new}}}(\theta) = \Big( \mathbf{M} + \alpha\beta \sum_{i=1}^{|\mathcal{T}|} \nabla_\theta \mathcal{L}_{\text{val}}^{\mathcal{T}_i}(\theta^{\tau_i}) \nabla_\theta \mathcal{L}_{\text{tr}}^{\mathcal{T}_i}(\theta)^\top \Big) \nabla_\theta \mathcal{L}_{\text{tr}}^{\mathcal{T}_{\text{new}}}(\theta) \tag{8}$$

$$= \mathbf{M} \nabla_\theta \mathcal{L}_{\text{tr}}^{\mathcal{T}_{\text{new}}}(\theta) + \beta \sum_{i=1}^{|\mathcal{T}|} \big( \nabla_\theta \mathcal{L}_{\text{tr}}^{\mathcal{T}_i}(\theta)^\top \nabla_\theta \mathcal{L}_{\text{tr}}^{\mathcal{T}_{\text{new}}}(\theta) \big) \alpha \nabla_\theta \mathcal{L}_{\text{val}}^{\mathcal{T}_i}(\theta^{\tau_i}) \tag{9}$$

$$= \mathbf{M} \nabla_\theta \mathcal{L}_{\text{tr}}^{\mathcal{T}_{\text{new}}}(\theta) + \beta \sum_{i=1}^{|\mathcal{T}|} \big( \underbrace{\nabla_\theta \mathcal{L}_{\text{tr}}^{\mathcal{T}_i}(\theta)^\top \nabla_\theta \mathcal{L}_{\text{tr}}^{\mathcal{T}_{\text{new}}}(\theta)}_{\text{A. Gradient similarity}} \big) \big( \underbrace{\alpha \nabla_\theta \mathcal{L}_{\text{val}}^{\mathcal{T}_i}(\theta) + \mathcal{O}(\alpha^2)}_{\text{B. Taylor expansion}} \big). \tag{10}$$

Given $\mathbf{M} = \mathbf{I}$, the second term in the R.H.S. of Eq. 10 can represent the final gradient direction for the new task. For Eq. 10, we used the Taylor expansion of vector-valued function, $\nabla_\theta \mathcal{L}_{\text{val}}^{\mathcal{T}_i}(\theta^{\tau_i}) \approx \nabla_\theta \mathcal{L}_{\text{val}}^{\mathcal{T}_i}(\theta) + \nabla_\theta^2 \mathcal{L}_{\text{val}}^{\mathcal{T}_i}(\theta)(\theta - \alpha \mathbf{M} \nabla_\theta \mathcal{L}_{\text{tr}}^{\mathcal{T}_i}(\theta) - \theta)$.

The term A of Eq. 10 is the inner product between the gradients of meta-training losses and new test losses. We can simply interpret this as how similar the gradient directions between two different tasks. This has been explicitly used in continual learning or multi-task learning setup to consider task similarity [7, 23, 36]. When we have a loss function in the form of finite sums, this term can be also interpreted as a kernel similarity between the respective sets of gradients (see Eq. 4 of [28]).

With the first term in B of Eq. 10, we compute a linear combination of the gradients of validation losses from the meta-training set. Its weighting factors are computed based on the similarities between the tasks from the meta-training set and the new task as explained above. Therefore, we essentially perform a soft nearest neighbor voting to find the direction among the validation gradients from the meta-training set. Given the new task, the gradient may lead the model to overfit (or underfit). However, the proposed method will extract the knowledge from the past experiences and find the gradients that gave us good validation performance during the meta-training process.

## 5 Related Work

**Meta-learning:** Model-agnostic meta-learning (MAML) highlighted the importance of the model's initial parameters for better generalization [10] and there have been many extensions to improve the framework, e.g. for continuous adaptation [1], better credit assignment [37], and robustness [15]. In this work, we improve the inner update optimizers by learning a curvature for better generalization and fast model adaptation. Meta-SGD [22] suggests to learn coordinate-wise learning rates. We can interpret it as an diagonal approximation to meta-curvature in a similar vein to recent adaptive learning rates methods, such as [43, 16, 8], performing diagonal approximations of second-order matrices. Recently, [4] suggested to learn layer-wise learning rates through the meta-training. However, both methods do not consider the dependencies between the parameters, which was crucial to provide more robust meta-training process and faster convergence. [21] also attempted to transform the gradients. They used simple binary mask applied to the gradient update to determine which parameters are to be updated while we introduce dense learnable tensors to model second-order dependencies with a series of tensor products.

**Few-shot classification:** As a good test bed to evaluate few-shot learning, huge progress has been made in the few-shot classification task. Triggered by [44], many recent studies have focused on discovering effective inductive bias on classification task. For example, network architectures that perform nearest neighbor search [44, 41] were suggested. Some improved the performance by modeling the interactions or correlation between training examples [26, 11, 42, 32, 29]. In order to overcome the nature of few-shot learning, the generative models have been suggested to augment the training data [40, 45] or generate model parameters for the specified task [39, 33]. The state-of-the-art results are achieved by additionally training 64-way classification task for pretraining [33, 39, 32]

Table 2: Few-shot classification results on Omniglot dataset. $^\dagger$ denotes 3 model ensemble.

| | 5-way 1-shot | 5-way 5-shot | 20-way 1-shot | 20-way 5-shot |
|---|---|---|---|---|
| SNAIL [27] | $99.07 \pm 0.16$ | $99.78 \pm 0.09$ | $97.64 \pm 0.30$ | $99.36 \pm 0.18$ |
| GNN [12] | 99.2 | 99.7 | 97.4 | 99.0 |
| MAML | $98.7 \pm 0.4$ | $99.9 \pm 0.1$ | $95.8 \pm 0.3$ | $98.9 \pm 0.2$ |
| Meta-SGD | $99.53 \pm 0.26$ | $99.93 \pm 0.09$ | $95.93 \pm 0.38$ | $98.97 \pm 0.19$ |
| MAML++$^\dagger$ [4] | 99.47 | 99.93 | $97.65 \pm 0.05$ | $99.33 \pm 0.03$ |
| MC1 | $99.47 \pm 0.27$ | $99.57 \pm 0.12$ | $97.60 \pm 0.29$ | $99.23 \pm 0.08$ |
| MC2 | $99.77 \pm 0.17$ | $99.79 \pm 0.10$ | $97.86 \pm 0.26$ | $99.24 \pm 0.07$ |
| MC2$^\dagger$ | $\mathbf{99.97 \pm 0.06}$ | $99.89 \pm 0.06$ | $\mathbf{99.12 \pm 0.16}$ | $\mathbf{99.65 \pm 0.05}$ |

with larger ResNet models [33, 39, 29, 26]. In this work, our focus is to improve the model-agnostic few-shot learner that is broadly applicable to other tasks, e.g. reinforcement learning setup.

**Learning optimizers:** Our proposed method may fall within the *learning optimizer* category [34, 3, 46, 25]. They also take as input the gradient and transform it via a neural network to achieve better convergence behavior. However, their main focus is to capture the training dynamics of individual gradient coordinates [34, 3] or to obtain a generic optimizer that is broadly applicable for different datasets and architectures [46, 25, 3]. On the other hand, we meta-learn a curvature coupled with the model's initialization parameters. We focus on a fast adaptation scenario requiring a small number of gradient steps. Therefore, our method does not consider a history of the gradients, which enables us to avoid considering a complex recurrent architecture. Finally, our approach is well connected to existing second order methods while learned optimizers are not easily interpretable since the gradient passes through nonlinear and multilayer recurrent neural networks.

# 6 Experiments

We evaluate the proposed method on a synthetic data few-shot regression task few-shot image classification tasks with Omniglot and MiniImagenet datasets. We test two versions of the meta-curvature. The first one, named as MC1, we fixed the $\mathbf{M}_o = \mathbf{I}$ Eq. 4. The second one, named as MC2, we learn all three meta-curvature matrices. We also report results on few-shot reinforcement learning in appendices.

## 6.1 Few-shot regression

To begin with, we perform a simple regression problem following [9, 22]. During the meta-training process, sinusoidal functions are sampled, where the amplitude and phase are varied within $[0.1, 5.0]$ and $[0, \pi]$ respectively. The network architecture and all hyperparameters are same as [9] and we only introduce the suggested meta-curvature. We reported the mean squared error with 95% confidence interval after one gradient step in Figure 1. The details are provided in appendices.

Table 1: Few-shot regression results.

| Method | 5-shot | 10-shot |
|---|---|---|
| MAML | $0.686 \pm 0.070$ | $0.435 \pm 0.039$ |
| Meta-SGD | $0.482 \pm 0.061$ | $0.258 \pm 0.026$ |
| LayerLR | $0.528 \pm 0.068$ | $0.269 \pm 0.027$ |
| MC1 | $0.426 \pm 0.054$ | $0.239 \pm 0.025$ |
| MC2 | $\mathbf{0.405 \pm 0.048}$ | $\mathbf{0.201 \pm 0.020}$ |

## 6.2 Few-shot classification on Omniglot

The Omniglot dataset consists of handwritten characters from 50 different languages and 1632 different characters. It has been widely used to evaluate few-shot classification performance. We follow the experimental protocol in [9] and all hyperparameters and network architecture are same as [9]. Further experimental details are provided in appendices. Except 5-shot 5-way setting, our simple 4 layers CNN with meta-curvatures outperform all MAML variants and also achieved state-of-the-art results without additional specialized architectures, such as attention module (SNAIL [27]) or relational module (GNN [12]). We provide the training curves in Figure 2 and our methods converge much faster and achieve higher accuracy.

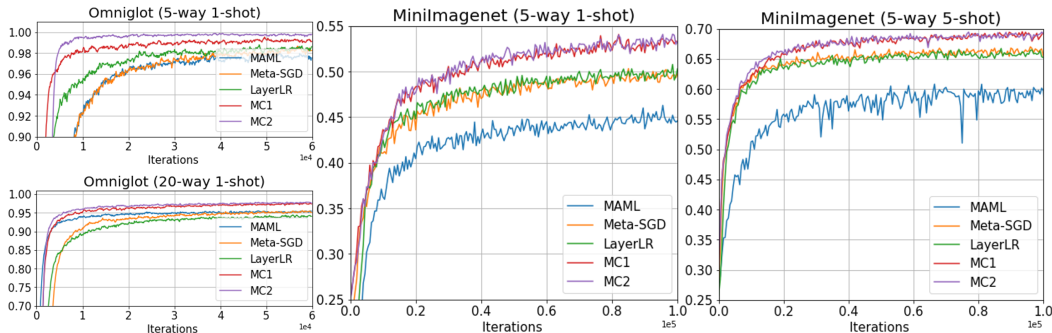

Figure 2: Few-shot classification accuracy over training iterations.

Table 3: Few-shot classification results on miniImagenet test set (5-way classification) with baseline 4 layer CNNs. * is from the original papers. [†] denotes 3 model ensembles.

| | 1-shot | | 5-shot | |
|---|---|---|---|---|
| Inner steps | 1 step | 5 step | 1 step | 5 step |
| *MAML | · | $48.7 \pm 1.84$ | · | $63.1 \pm 0.92$ |
| *Meta-SGD | $50.47 \pm 1.87$ | · | $64.03 \pm 0.94$ | · |
| *MAML++[†] | $51.05 \pm 0.31$ | $52.15 \pm 0.26$ | · | $68.32 \pm 0.44$ |
| MAML | $46.28 \pm 0.89$ | $48.85 \pm 0.88$ | $59.26 \pm 0.72$ | $63.92 \pm 0.74$ |
| Meta-SGD | $49.87 \pm 0.87$ | $48.99 \pm 0.86$ | $66.35 \pm 0.72$ | $63.84 \pm 0.71$ |
| LayerLR | $50.04 \pm 0.87$ | $50.55 \pm 0.87$ | $65.06 \pm 0.71$ | $66.64 \pm 0.69$ |
| MC1 | $53.37 \pm 0.88$ | $53.74 \pm 0.84$ | $\mathbf{68.47 \pm 0.69}$ | $\mathbf{68.01 \pm 0.73}$ |
| MC2 | $\mathbf{54.23 \pm 0.88}$ | $\mathbf{54.08 \pm 0.93}$ | $67.94 \pm 0.71$ | $67.99 \pm 0.73$ |
| MC2[†] | $\mathbf{54.90 \pm 0.90}$ | $\mathbf{55.73 \pm 0.94}$ | $\mathbf{69.46 \pm 0.70}$ | $\mathbf{70.33 \pm 0.72}$ |

### 6.3 Few-shot classification on miniImagenet and tieredImagenet

**Datasets:** The miniImagenet dataset was proposed by [44, 34] and it consists of 100 subclasses out of 1000 classes in the original dataset (64 training classes, 12 validation classes, 24 test classes). The tieredImagenet dataset [35] is a larger subset, composed of 608 classes and reduce the semantic similarity between train/val/test splits by considering high-level categories.

**baseline CNNs:** We used 4 layers convolutional neural network with the batch normalization followed by a fully connected layer for the final classification. In order to increase the capacity of the network, we increased the filter size up to 128. We found that the model with the larger filter seriously overfit (also reported in [9]). To avoid overfitting, we applied data augmentation techniques suggested in [5, 6]. For a fair comparison to [4], we also reported the results of model ensemble. Throughout the meta-training, we saved the model regularly and picked 3 models that have the best accuracy on the meta-validation dataset. We re-implemented all three baselines and performed the experiments with the same settings. We provide further the details in the appendices.

Fig. 2 and Table 3 shows the results of baseline CNNs experiments on miniImagenet. MC1 and MC2 outperformed all other baselines for all different experiment settings. Not only does MC reach a higher accuracy at convergence, but also showed a much faster convergence rates for meta-training. Our methods share the same benefits as second order methods although we do not approximate any Hessian or Fisher matrices. Unlike other MAML variants, which required an extensive hyperparameter search, our methods are very robust to hyperparameter settings. Usually, MC2 outperforms MC1 because the more fine-grained meta-curvature enable us to effectively increase the model's capacity.

**WRN-28-10 features and MLP:** To the best of our knowledge, [39, 33] are current state-of-the-art methods that use a pretrained WRN-28-10 [47] network (trained with 64-way classification task on entire meta-training set) as a feature extractor network. We evaluated our methods on this setting by adding one hidden layer MLP followed by a softmax classifier and our method again improved MAML variants by a large margin. Despite our best attempts, we could not find a good hyperparameters to

Table 4: The results on miniImagenet and tieredImagenet. ‡ indicates that both meta-train and meta-validation are used during meta-training. † denotes indicates that 15-shot meta-training was used for both 1-shot and 5-shot testing. MetaOptNet [3] used ResNet-12 backbone and trained end-to-end manner while we used the fixed features provided by [2] (center - features from the central crop, multiview - features averaged over four corners, central crops, and horizontal mirrored).

| | miniImagenet | | tieredImagenet | |
|---|---|---|---|---|
| | 1-shot | 5-shot | 1-shot | 5-shot |
| [33]‡ | $59.60 \pm 0.41$ | $73.74 \pm 0.19$ | $\cdot$ | $\cdot$ |
| LEO (center)‡ [39] | $61.76 \pm 0.08$ | $77.59 \pm 0.12$ | $66.33 \pm 0.05$ | $81.44 \pm 0.09$ |
| LEO (multiview)‡ [39] | $63.97 \pm 0.20$ | $79.49 \pm 0.70$ | $\cdot$ | $\cdot$ |
| MetaOptNet-SVM‡† [20] | $64.09 \pm 0.62$ | $80.00 \pm 0.45$ | $65.81 \pm 0.74$ | $81.75 \pm 0.53$ |
| Meta-SGD (center) | $56.58 \pm 0.21$ | $68.84 \pm 0.19$ | $59.75 \pm 0.25$ | $69.04 \pm 0.22$ |
| MC2 (center) | $61.22 \pm 0.10$ | $75.92 \pm 0.17$ | $66.20 \pm 0.10$ | $82.21 \pm 0.08$ |
| MC2 (center)‡ | $\mathbf{61.85 \pm 0.10}$ | $77.02 \pm 0.11$ | $\mathbf{67.21 \pm 0.10}$ | $\mathbf{82.61 \pm 0.08}$ |
| MC2 (multiview)‡ | $\mathbf{64.40 \pm 0.10}$ | $\mathbf{80.21 \pm 0.10}$ | $\cdot$ | $\cdot$ |

train original MAML in this setting. Although our main goal is to push how much a simple gradient transformation in the inner loop optimization can improve general and broadly applicable MAML frameworks, our methods outperformed the recent methods that used various task specific techniques, e.g. task dependent weight generating methods [39, 33] and relational networks [39]. Our methods also outperformed the very latest state of the art results [20] that used extensive data-augmentation, regularization, and 15-shot meta-training schemes with different backbone networks.

## 7 Conclusion

We propose to meta-learn the curvature for faster adaptation and better generalization. The suggested method significantly improved the performance upon previous MAML variants and outperformed the recent state of the art methods. It also leads to faster convergence during meta-training. We present an analysis about generalization performance and connect to existing second order methods, which would provide useful insights for further research.

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
