[Supplementary Material · meta_curvature_supp.pdf]

# Meta-Curvature: Supplementary Materials

## 1 Meta-training algorithm

Alg. 1 shows the details of the algorithm to train meta-curvature matrices and the initial model parameters. To avoid cluttered notation, we assumed the model has only one layer and it is straightforward to extend to multiple layers.

## 2 Case study

In this section, we provide a case study of the linear regression example. Let $\mathbf{X}_{\text{tr}}, \mathbf{X}_{\text{val}}, \mathbf{X}_{\text{new}} \in \mathbb{R}^{m \times p}$ training, validation, and test set and their targets are $\mathbf{Y}_{\text{tr}}, \mathbf{Y}_{\text{val}}, \mathbf{Y}_{\text{new}} \in \mathbb{R}^{m}$. With the model's parameter $\theta \in \mathbb{R}^{p}$, a typical loss function for the linear regression is defined as follows.

$$J(\theta) = \frac{1}{2}\|\mathbf{Y} - \mathbf{X}\theta\|^2. \tag{1}$$

The gradient w.r.t the model's parameter $\theta$ is

$$\nabla_\theta J(\theta) = -\mathbf{X}^\top(\mathbf{Y} - \mathbf{X}\theta). \tag{2}$$

Given the meta-curvature matrix, $\mathbf{M}$, a fixed inner learning rate $\alpha$, then the meta-objective function is

$$J_{\text{val}}(\theta) = \frac{1}{2}\|\mathbf{Y}_{\text{val}} - \mathbf{X}_{\text{val}}(\theta^{\text{tr}})\|^2 \tag{3}$$

Following the derivation from the main text and given the new test set, we perform one inner and outer optimization steps. And the transformed gradient for the new test set is as follow.

$$\mathbf{M}_{\text{new}}\nabla_\theta J_{\text{new}}(\theta) \tag{4}$$

$$= \mathbf{M}\nabla_\theta J_{\text{new}}(\theta) + \beta\big(\nabla_\theta J_{\text{tr}}(\theta)^\top \nabla_\theta J_{\text{new}}(\theta)\big)\alpha\nabla_\theta J_{\text{val}}(\theta^{\text{tr}}) \tag{5}$$

$$= \mathbf{M}\nabla_\theta J_{\text{new}}(\theta) - \beta\big[(\mathbf{Y}_{\text{tr}} - \mathbf{X}_{\text{tr}}\theta)^\top \mathbf{X}_{\text{tr}}\mathbf{X}_{\text{new}}^\top(\mathbf{Y}_{\text{new}} - \mathbf{X}_{\text{new}}\theta)\big]\alpha\nabla_\theta J_{\text{val}}(\theta^{\text{tr}}) \tag{6}$$

$$= \mathbf{M}\nabla_\theta J_{\text{new}}(\theta) - \beta\big[(\mathbf{Y}_{\text{tr}} - \mathbf{X}_{\text{tr}}\theta)^\top \mathbf{X}_{\text{tr}}\mathbf{X}_{\text{new}}^\top(\mathbf{Y}_{\text{new}} - \mathbf{X}_{\text{new}}\theta)\big]\alpha\big[\mathbf{X}_{\text{val}}^\top(\mathbf{Y}_{\text{val}} - \mathbf{X}_{\text{val}}\theta^{\text{tr}})\big] \tag{7}$$

$$= \mathbf{M}\nabla_\theta J_{\text{new}}(\theta) - \beta\big[(\mathbf{Y}_{\text{tr}} - \mathbf{X}_{\text{tr}}\theta)^\top \mathbf{X}_{\text{tr}}\mathbf{X}_{\text{new}}^\top(\mathbf{Y}_{\text{new}} - \mathbf{X}_{\text{new}}\theta)\big]$$
$$\alpha\big[\mathbf{X}_{\text{val}}^\top\mathbf{Y}_{\text{val}} - \mathbf{X}_{\text{val}}^\top\mathbf{X}_{\text{val}}(\theta - \alpha\mathbf{M}\mathbf{X}_{\text{tr}}^\top(\mathbf{Y}_{\text{tr}} - \mathbf{X}_{\text{tr}}\theta))\big] \tag{8}$$

$$= \mathbf{M}\nabla_\theta J_{\text{new}}(\theta) - \beta\big[\underbrace{(\mathbf{Y}_{\text{tr}} - \mathbf{X}_{\text{tr}}\theta)^\top \mathbf{X}_{\text{tr}}\mathbf{X}_{\text{new}}^\top(\mathbf{Y}_{\text{new}} - \mathbf{X}_{\text{new}}\theta)}_{\text{A}}\big]$$
$$\big[\alpha\underbrace{\mathbf{X}_{\text{val}}^\top(\mathbf{Y}_{\text{val}} - \mathbf{X}_{\text{val}}\theta)}_{\text{B}} - \alpha^2\underbrace{\mathbf{X}_{\text{val}}^\top\mathbf{X}_{\text{val}}}_{\text{C}}\underbrace{\mathbf{M}\mathbf{X}_{\text{tr}}^\top(\mathbf{Y}_{\text{tr}} - \mathbf{X}_{\text{tr}}\theta)}_{\text{D}}\big]. \tag{9}$$

The term A is the gradient similarity term, and in linear regression case, it is defined as a bilinear form e.g. $\mathbf{x}^\top\mathbf{A}\mathbf{y}$, where $\mathbf{A} = \mathbf{X}_{\text{tr}}\mathbf{X}_{\text{new}}^\top$. It is multiplied by both training and test residuals. $\mathbf{A}$ is related to

**Algorithm 1** Training MAML with the meta-curvature for few-shot supervised learning

---
**Input:** task distribution $p(\mathcal{T})$, learning rate $\alpha, \beta$
Initialize $\mathbf{M}_o, \mathbf{M}_i, \mathbf{M}_f = \mathbf{I}$
**while** not converged **do**
    Sample batch of tasks $\tau_i \sim p(\mathcal{T})$
    **for all** $\tau_i$ **do do**
        $\theta^{\tau_i} = \theta - \alpha \mathbf{M}_{mc} \nabla \mathcal{L}_{\text{tr}}^{\tau_i}(\theta)$ {Assuming one gradient step}
    **end for**
    $\theta \leftarrow \text{ADAM}\big(\theta, \beta, \nabla_\theta \sum_{\tau_i} \mathcal{L}_{\text{val}}^{\tau_i}(\theta^{\tau_i})\big)$
    $\mathbf{M}_o \leftarrow \text{ADAM}\big(\mathbf{M}_o, \beta, \nabla_{\mathbf{M}_o} \sum_{\tau_i} \mathcal{L}_{\text{val}}^{\tau_i}(\theta^{\tau_i})\big)$
    $\mathbf{M}_i \leftarrow \text{ADAM}\big(\mathbf{M}_i, \beta, \nabla_{\mathbf{M}_i} \sum_{\tau_i} \mathcal{L}_{\text{val}}^{\tau_i}(\theta^{\tau_i})\big)$
    $\mathbf{M}_f \leftarrow \text{ADAM}\big(\mathbf{M}_f, \beta, \nabla_{\mathbf{M}_f} \sum_{\tau_i} \mathcal{L}_{\text{val}}^{\tau_i}(\theta^{\tau_i})\big)$
**end while**

---

Table 1: Few-shot regression results on sinusoidal functions.

| Method | 5-shot | 10-shot | 20-shot |
|---|---|---|---|
| MAML | $0.686 \pm 0.070$ | $0.435 \pm 0.039$ | $0.228 \pm 0.024$ |
| Meta-SGD | $0.482 \pm 0.061$ | $0.258 \pm 0.026$ | $0.127 \pm 0.013$ |
| LayerLR | $0.528 \pm 0.068$ | $0.269 \pm 0.027$ | $0.134 \pm 0.014$ |
| MC1 | $0.426 \pm 0.054$ | $0.239 \pm 0.025$ | $0.125 \pm 0.013$ |
| MC2 | $\mathbf{0.405 \pm 0.048}$ | $\mathbf{0.201 \pm 0.020}$ | $\mathbf{0.112 \pm 0.011}$ |

covariance matrix, but between training set and the new test set. The term B is the validation gradient term. The terms C and D correspond to $\mathcal{O}(\alpha^2)$. Since the loss function of the linear regression has a quadratic form and its derivative has a linear form. Therefore, the Taylor expansion of the derivative has up to $\alpha^2$ order. The term D is the transformed gradient and the term C is a covariance matrix of validation dataset (assuming it's centered).

# 3   Few-shot regression

**Experimental setup** We used the same experimental setups in [1]. During training and testing, the amplitude and the phase vary within $[0.1, 5.0]$ and $[0, \pi]$ respectively, and data points are sampled from uniform distribution $[-5, 5]$. We used one gradient step with the fixed learning rate 0.01 and Adam was used for meta-training with the outer loop learning rate 0.001. We used the same network architecture, which has two 40 dimension fully connected layers with ReLU activation. We sampled 25 tasks for every iterations and trained 70000 iterations. We reported the performance from the trained model that had the minimum loss value. [1] reported the MSE for 5-shot setting, and we could reproduced the results. [2] has slightly different settings, so the MSE are not directly comparable to theirs.

**Qualitative results**: We provide qualitative results of few-shot regression task on sinusoidal functions in Figure 1. The star shape markers are the few data points for training, and we draw the curves based on each methods, MAML, Meta-SGD, and the proposed MC2. The left column is 5-shot and the right column is 10-shot experiments.

# 4   Few-shot classification on Omniglot dataset

We used the same experimental setups in [1]. Out of 1623 characters, we used 1100 characters for training, 100 characters for validation, and remaining 423 characters for testing. The network architecture is 4 convolutional layers with 64 filters and 1 fully connected layer for the final classification. We only used one inner gradient step with 0.4 learning rate for all meta-curvature experiments for training and testing. The batch size was set to 32 (5-way) and 16 (20-way), and outer loop learning rate is 0.001 and we trained 60000 iterations.

# 5 Few-shot classification on miniImagenet and tieredImagenet dataset

## 5.1 baseline CNNs

For both 5-way 1-shot and 5-way 5-shot classification, we set the batch size 4 for 1 step experiments and 2 for 5 step experiments. 15 examples per class were used for evaluating the model after updates. In total, we ran 100,000 iterations for 1 step experiments and 200,000 iterations for 2 step experiments. The inner/outer learning rates are $\beta = 0.001, \alpha = 0.01$. We apply dropout rate 0.2 in the final linear layer for only MC1 and MC2 (other methods did perform worse with dropout). For cutout data augmentation, we cut out $36 \times 36$ random crops.

## 5.2 WRN-28-10 features and MLP

We used the WRN-28-10 features provided by [4]. Although they also provide multi-view features (average of center and corner crops), we used a feature from the image center. The dimension of feature was 640 and we used one hidden layer (2048 units, ReLU activation function) followed by a softmax classifier. We used one gradient step for 1-shot experiments and 5 steps for 5-shot experiments. For 5-shot experiments, we used separate meta-curvature matrices for each iterations. Every training iterations, we sample random 16 tasks, and set the initial learning rate 0.005 and decay the learning rate down to 0.0001 by using a cosine annealing schedule. And we trained 30000 iterations and used early stopping scheme to report the final performance.

# 6 Few-shot reinforcement learning

The goal of few-shot learning in reinforcement learning (RL) is that an agent can quickly adapt to a new task with little prior experience. A distinct feature from the few-shot supervised learning task is that the RL objective is not generally differentiable. Therefore, we use policy gradient methods to estimate the gradient both for inner and outer loop gradients. In addition, policy gradient methods are generally on-policy, which means that the training data depends on the agent's initial policy. Therefore, the initial policy (with the meta-learned initial parameters) needs to explore as diverse experiences as possible to get proper feedback from a new task. We described the method and interpretation with respect to supervised classification tasks, but it can be easily modified to RL setting.

## 6.1 Experimental setup

We tested our method on complex high-dimensional locomotion tasks with the MuJoCo simulator [7]. Most of the settings are based on [1] for fair comparison. We consider two simulated robots (HalfCheetah and Walker2d) and two types of task environments (to run in a forward/backward direction or a particular velocity). The network architecture is two hidden layers of size 100 with ReLU activations for both. We used the standard linear feature baseline estimator. We evaluated the performance after one policy gradient step with 20 trajectories. We compare against MAML-TRPO and MAML-PPO. In the original MAML, TRPO [5] was used as the outer loop optimizer but we found out that using PPO [6] consistently outperformed the TRPO. MAML-PPO is also computationally more efficient since MAML-TRPO requires third-order gradients (or computed by hessian-vector product instead). To the best of our knowledge, MAML-PPO has not been tested on this setup. We evaluated two variations of meta-curvature similar to the classification setup, MC1 and MC2, and used PPO as the meta-optimizer. Note that this is a preliminary result, so this is not by no means conclusive. We provide this information for the readers who might be interested in this direction.

## 6.2 Experimental results

Fig. 2 shows the rewards obtained after one step policy gradient update. In the HalfCheetahDir experiment, our methods outperformed both strong baselines. MC1-PPO reached the same performance of a strong baseline, MAML-PPO three times faster. In HalfCheetahVel and Walker2dDir experiments, both MC2-PPO and MAML-PPO reached nearly the same performance, but in a more sample efficient manner. For Walker2dVel, MAML-TRPO showed the fastest convergence at the

earlier meta-training stage, but our meta-curvatures outperformed eventually. In this setting, most of the rewards come from the survival reward (the agent gets $1.0$ reward for every step if they do not fall over). All methods were able to survive throughout the episode, but our methods run better at a given velocity. One thing we noticed that it stops obtaining more rewards and starts to degrade the performance in Walker2dDir experiment. The recently proposed approach [3] may alleviate this issue through better credit assignment in the meta-gradients. Combining it would be interesting direction to be explored.

# 7  Visualization

Fig. 3 is a visualization of meta-trained meta-curvature matrices for 5-way 1-shot classification task. To visualize the full matrix, $\mathbf{M}_{mc}$, we picked up the matrices from the first convolutional layer in the small model (filter size 64). Therefore with the 3 color input channels, $\mathbf{M}_f \in \mathbb{R}^{9 \times 9}$, $\mathbf{M}_i \in \mathbb{R}^{3 \times 3}$, $\mathbf{M}_o \in \mathbb{R}^{64 \times 64}$, and $\mathbf{M}_{mc} \in \mathbb{R}^{1728 \times 1728}$. The diagonal elements are high values, mostly $> 0.5$. Interestingly, there are also a lot of off-diagonal elements $> 0.5$ or $< -0.5$. Thus, they capture the dependencies between the gradients.

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

Figure 1: Qualitative results of few-shot regression on sinusoidal functions. The left column - 5 shot, The right column - 10 shot

Figure 2: Reinforcement learning experimental results. Y-axis: rewards after the model updates. X-axis: meta-training steps. We performed at least three runs with random seeds and the curves are averaged over them.

Figure 3: Visualization of meta-curvature matrices. We clipped the values $[-1, 1]$ for better visualization (Best viewed in color)