[Reviews · NeurIPS 2019]

Reviewer 1



Overall opinion This is a good paper. The motivation is clear and the method is simple. The paper is meaningful in that it provides a new method to parameterize and learn a curvature in weight space. The analysis makes connections to second-order methods and also shows how the learned curvature operates in terms of previous gradients. Experiments show the advantages of their method. +Clearly written. I especially appreciated the crisp introduction to tensor algebra (section 2.1). +Analysis shows connections to Fisher information matrix, and the decomposition of the altered gradient (eq 10) shows how this method implicitly “memorizes” previous train- and val- gradients. +Experiments show that meta-curvature outperforms MAML, meta-SGD, MAML++, and layerLR -WRN experiments (Table 4) show that meta-curvature is slightly outperformed by LEO, though I believe this can be overlooked as LEO results involved heavy engineering. -The paper did not cite and compare to a relevant previous work with a similar motivation: [1] also learns additional parameters for MAML to learn to alter the gradient, and the learned parameters correspond to a learned curvature for each layer. Minor comments: -line 75: If I understand correctly, it is n-mode unfolding, matrix multiplication, and then reverse(?) n-mode unfolding. Otherwise, the result would be a 2-order rather than an N-order tensor. -section 3.2.2: I personally found this, along with figure 1, harder to follow than section 3.2.1. [1] Gradient-Based Meta-Learning with Learned Layerwise Metric and Subspace, ICML 2018 ------------------------ post-rebuttal ---------------------------- The authors have addressed all of my concerns, and the additional experiments highlight the advantage of meta-curvature even more. I maintain my score.

Reviewer 2



This submission aims to meta learn curvature estimations such that it will lead to better generalization than Hessian or Fisher-information matrix. In terms of writing, this work is well written. A concern is I couldn’t find how is equation (10) leads to better generalization, since the second term in equation (10) is a weighted gradient on validation dataset, the weights reflect the similarity between gradient of training loss and test loss but do not link to generalization directly.

Reviewer 3



Originality: The paper is quite original, I am not aware of similar studies. Clarity: Overall, the paper is written well, especially the most challenging sections 3.2 and 4. I have a number of targeted comments thought that I think should be addressed before publication: 1. Line 34: “They compute local curvatures with training losses and move along the curvatures as far as possible.”

I am not sure what authors mean here, perhaps some clarification would be helpful. 2. Line 130: Why is the layer parametrised in this way? If it is a convolutional layer, I would expect a 4rd dimension, the number of output channels. Do authors also consider fully-connected layers? What kind of curvature parametrisation is supposed to be used in that case? 3. Can authors provide more comments on the representational power of their _parametrizations_ (not gradients), e.g. with comparison to other tensor methods, especially Tensor-Train decomposition? 4. Does the batch normalization anyhow affect the analysis provided in section 4 due to gradient sharing? Can we get rid of the batch norm at all as the learned curvature can learn even a better normalisation scheme? 5. I have got a bit confused with the “train”, “validation” and “test” sets used in section 4, in the standard MAML setup is the meta-update computed on the “train” set and the initialisation is updated based on the loss on the “validation” set (using the paper’s terminology)? If so, I am not sure if the “validation” is the best term to use. Quality: The proposed method seems to be solid. The experimental comparison is also broad enough to claim an empirical contribution. Significance: The problem of generalisation in few-shot learning is very actual and the paper addresses this problem in a novel, interesting way. Learned curvatures might find applications in other problems too.

[Author Response · NeurIPS 2019]

## 1 Rebuttal - Meta-Curvature (Paper ID: 1842)

## 2 To all reviewers:

We performed more hyperparameter search, e.g. dropout rates, learning rates, the number of hidden units, etc., and the results had shown that the proposed method outperformed LEO [2] and the recent state-of-the-art results from [3]. For fair comparison to [3] that used extensive data augmentations, we also reported the results with the multiview features provided by LEO [2], where features were averaged over representations of 4 corner and central crops and their horizontal mirrored versions. We did not make any algorithmic changes and we are hoping to update the results with the corresponding hyperparameters in the final version. We promise to release the code and trained models in order to encourage reproducibility.

Table 1: The results on miniImagenet and tieredImagenet with WRN-28-10 features. $^{\ddagger}$ indicates that both meta-train and meta-validation are used during meta-training. $^{\dagger}$ denotes indicates that 15-shot meta-training was used for both 1-shot and 5-shot testing. MetaOptNet [3] used ResNet-12 backbone and trained end-to-end manner while we used the fixed features provided by [2].

|  | miniImagenet | | tieredImagenet | |
|---|---|---|---|---|
|  | 1-shot | 5-shot | 1-shot | 5-shot |
| [1]$^{\ddagger}$ | $59.60 \pm 0.41$ | $73.74 \pm 0.19$ | · | · |
| LEO (center)$^{\ddagger}$ [2] | $61.76 \pm 0.08$ | $77.59 \pm 0.12$ | $66.33 \pm 0.05$ | $81.44 \pm 0.09$ |
| LEO (multiview)$^{\ddagger}$ [2] | $63.97 \pm 0.20$ | $79.49 \pm 0.70$ | · | · |
| MetaOptNet-SVM$^{\ddagger\dagger}$ [3] | $64.09 \pm 0.62$ | $80.00 \pm 0.45$ | $65.81 \pm 0.74$ | $81.75 \pm 0.53$ |
| Meta-SGD (center) | $56.58 \pm 0.21$ | $68.84 \pm 0.19$ | $59.75 \pm 0.25$ | $69.04 \pm 0.22$ |
| MC2 (center) | $61.22 \pm 0.10$ | $75.92 \pm 0.17$ | $66.20 \pm 0.10$ | $82.21 \pm 0.08$ |
| MC2 (center)$^{\ddagger}$ | $\mathbf{61.85 \pm 0.10}$ | $77.02 \pm 0.11$ | $\mathbf{67.21 \pm 0.10}$ | $\mathbf{82.61 \pm 0.08}$ |
| MC2 (multiview)$^{\ddagger}$ | $\mathbf{64.40 \pm 0.10}$ | $\mathbf{80.21 \pm 0.10}$ | · | · |

## 11 To R1:

Thanks for the reference, we will cite it with the discussion. In short, the big difference is that their matrix to transform the gradient is a simple binary mask whose rows are either 0 or 1 vector. With the updated experimental results, we hope we resolve your concerns about the performance of WRN experiments.

## 16 To R2:

Eq (10): Given a new task, it does not directly follow the gradients of training loss, which might lead the model to overfit (or underfit). Instead, it finds the most similar tasks in the meta-training set and follows the gradients of validation losses in those similar tasks.

## 20 To R3:

For clarity question 1: The second order optimization methods are mainly for speeding up the convergence. However, there is no notion of generalization. Faster convergence could mean faster overfitting, which may lose the opportunity to get out of local minima.

2: In convolutional layers, we collapsed height and width into one dimension. The filter size is usually very small (3x3), the second-order matrix (9x9) might not be a big issue. In fully-connected layers, for example, a weight matrix 10x20 needs two curvature matrices, 10x10 and 20x20.

3-4: We do really appreciate your comments about tensor-train decomposition and batch normalization. Both are really interesting aspects. We will leave them as future works.

5: We agree with your point. Here, 'test' set is probably better choice than 'validation' set.

## 30 References

[1] Siyuan Qiao, Chenxi Liu, Wei Shen, Alan Yuille, Few-Shot Image Recognition by Predicting Parameters from Activations, In *CVPR* 2018.

[2] Andrei A. Rusu, Dushyant Rao, Jakub Sygnowski, Oriol Vinyals, Razvan Pascanu, Simon Osindero, Raia Hadsell, Meta-Learning with Latent Embedding Optimization, In *ICLR* 2019.

[3] Kwonjoon Lee, Subhransu Maji, Avinash Ravichandran, Stefano Soatto, Meta-Learning with Differentiable Convex Optimization, In *CVPR* 2019.



[Meta-Review · NeurIPS 2019]

This paper addresses a method for learning meta-curvature information from various similar tasks, which is claimed to help better generalization and fast model adaptation. A tensor decomposition is applied to the meta-curvature to scale up the method. All of reviewers agree that the paper is well written and the idea on the tensor decomposition of the curvature is interesting. However most of reviewers did not pinpoint a few issues. First of all, it is not clear why the method provides better generalization. Second, the meta-curvature in this paper is similar to the idea of he MT-net [Lee and Choi, 2018], while the novelty here is the application of tensor decomposition for scaling up. During the discussion period, unfortunately none of reviewers did express further feedback. There was no "strong support" or "negative support" either. Anyway, the paper is slightly above the threshold, deserved to be presented at NeurIPS this year.